# Orthopedic Treatment for Class II Malocclusion with Functional Appliances and Its Effect on Upper Airways: A Systematic Review with Meta-Analysis

**DOI:** 10.3390/jcm9123806

**Published:** 2020-11-25

**Authors:** Darius Bidjan, Rahel Sallmann, Theodore Eliades, Spyridon N. Papageorgiou

**Affiliations:** Clinic of Orthodontics and Pediatric Dentistry, Center of Dental Medicine, University of Zurich, 8032 Zurich, Switzerland; darius.bidjan@bluewin.ch (D.B.); rahel.sallmann@hotmail.com (R.S.); theodore.eliades@zzm.uzh.ch (T.E.)

**Keywords:** Class II malocclusion, mandibular retrognathism, orthopedic treatment, dentofacial orthopedics, orthodontics, functional appliances, clinical trials, systematic review, meta-analysis

## Abstract

Aim of this systematic review was to assess the effects of orthopedic treatment for Class II malocclusion with Functional Appliances (FAs) on the dimensions of the upper airways. Eight databases were searched up to October 2020 for randomized or nonrandomized clinical studies on FA treatment of Class II patients with untreated control groups. After duplicate study selection, data extraction, and risk of bias assessment according to Cochrane guidelines, random effects meta-analyses of mean differences (MDs) and their 95% confidence intervals (CIs) were performed, followed by subgroup/meta-regression analyses and assessment of the quality of evidence. A total of 20 nonrandomized clinical studies (4 prospective/16 retrospective) including 969 patients (47.9% male; mean age 10.9 years) were identified. Orthopedic treatment with FAs was associated with increased oropharynx volume (MD = 2356.14 mm^3^; 95% CI = 1276.36 to 3435.92 mm^3^; *p* < 0.001) compared to natural growth. Additionally, significant increases in nasopharynx volume, minimal constricted axial area of pharyngeal airway, and airway were seen, while removable FAs showed considerably greater effects than fixed FAs (*p* = 0.04). Finally, patient age and treatment duration had a significant influence in the effect of FAs on airways, as had baseline matching and sample size adequacy. Clinical evidence on orthopedic Class II treatment with FAs is associated with increased upper airway dimensions. However, the quality of evidence is very low due to methodological issues of existing studies, while the clinical relevance of increases in airway dimensions remains unclear.

## 1. Introduction

### 1.1. Background

Skeletal Class II malocclusion is the most common clinical entity the orthodontist is faced with [1] and is often due to a retrognathic mandible [2]. Among growing patients with a retrognathic mandible, orthopedic advancement of the mandible and its dentition with functional appliances is often performed with considerable success. However, functional appliances are now believed to have mostly dentoalveolar effects [3,4] and more limited effects on skeletal components [5,6,7].

At the same time, severe mandibular retrognathism has been linked to obstructive sleep apnea (OSA) [8] due to a retrodisplacement of the tongue and hyoid bone that may lead to a concomitant upper airway constriction [9,10]. Inversely, therapeutic advancement of the mandible with functional appliances among OSA patients has been shown to be an effective means to improve clinical OSA parameters [11]. Therefore, it might be reasonable to expect that functional appliance therapy among patients with skeletal Class II malocclusion might be associated with a beneficial effect on the patient’s airways [12] and possibly breathing function [13].

A previous systematic review on the subject [14] concluded that early treatment with functional appliances had positive effects on the upper airway, especially on oropharyngeal dimensions, in growing skeletal Class II patients and might decrease potential risk of OSA for growing patients in the future. However, this review only covered literature published only up to the start of 2017, while its conclusions might be influenced by existing methodological issues like lack of an a priori protocol [15], incomplete handling of risk of bias within studies according to the latest Cochrane guidelines [16], issues with the data synthesis (double-counting of controls from multiarm studies, outdated statistical modelling, lack of sensitivity analyses) [17], and no assessment of the quality of meta-evidence [18]. Finally, that systematic review only assessed overall effects, did not associate them with differences between removable/fixed appliances [3,4] and did not assess any patient risk factors.

### 1.2. Objective

Therefore, the aim of this systematic review was to compare the effects of functional appliance treatment for Class II malocclusion on the upper airway dimensions with natural occurring growth in untreated Class II patients.

## 2. Materials and Methods

### 2.1. Protocol, Registration, and Review Question

This review’s protocol was made a priori, registered in PROSPERO (CRD42019125897) with all post hoc changes transparently reported (Appendix B). The conduct and reporting of this review are guided by the Cochrane Handbook [19] and the PRISMA statement [20], respectively. The focused question this review tried to answer is: “Does functional appliance therapy of growing Class II patients lead to an increase in the upper airway dimensions to a degree greater than expected by natural growth alone?”.

### 2.2. Eligibility Criteria

Based on the Participants-Intervention-Comparison-Outcome-Study design (PICOS) schema and as few randomized trials exist on this matter, included were randomized and nonrandomized clinical studies on systemically healthy growing human patients of any age (<18 years), sex, and ethnicity with Class II malocclusion with mandibular retrognathism receiving orthopedic functional appliance treatment without any limitations on language, publication year, or status. Excluded were nonclinical studies, animal studies, and case reports/series, as well as studies with obstructive sleep apnea patients, studies without functional appliance treatment, and studies without an untreated longitudinal Class II control group. The primary outcome for this review was the total volume of the upper airways or any specific airway compartment assessed with Cone Beam Computerized Tomography (CBCT). Secondary outcomes included other measures of airway dimensions in terms of linear distances or areas, measured either on lateral cephalograms or CBCTs and in either upright or supine position.

### 2.3. Information Sources and Search

Eight electronic databases were searched without restrictions from inception to 20 October 2020 (Appendix A), while ClinicalTrials.gov Directory of Open Access Journals, Digital Dissertations, metaRegister of Controlled Trials, WHO, Google Scholar, and the reference/citation lists of included articles or existing systematic reviews were manually searched.

### 2.4. Study Selection

Two authors (D.B. and R.S.) screened the titles and/or abstracts of search hits to exclude obviously inappropriate studies, prior to checking their full texts. Any differences between the two reviewers were resolved by discussion with the last authors (T.E. and S.N.P.).

### 2.5. Data Collection Process and Items

Data from included studies was collected independently by two authors (D.B. and R.S.) with the same way to resolve discrepancies using predefined/piloted forms covering: (a) study characteristics (design, clinical setting, and country), (b) patient characteristics (age and sex), (c) eligibility criteria for patient selection, (d) treatment details (appliance and duration), and (e) outcome measurement modality.

### 2.6. Risk of Bias of Individual Studies

The risk of bias of included nonrandomized studies was assessed with a custom tool based on the ROBINS-I (“Risk Of Bias In Nonrandomized Studies—of Interventions”) [16]. Assessment of the risk of bias was likewise independently performed by two authors (DB, RS) with the same approach being applied to resolve discrepancies.

### 2.7. Data Synthesis and Summary Measures

An effort was made to maximize data for the analysis; where data were missing, they were calculated by ourselves. As the outcome of upper airway dimensions is bound to be affected by patient and treatment-related characteristics (baseline dimensions, growth potential, compliance, and response to treatment), a random-effects model was a priori deemed appropriate to calculate the average distribution of true effects, based on clinical and statistical reasoning [21], and a restricted maximum likelihood variance estimator with improved performance was used according to recent guidance [22]. Mean differences (MDs) with their corresponding 95% confidence intervals (CIs) were used, while the standardized mean difference (SMD) was decided post hoc to combine similar measurements of nasopharyngeal volume (Appendix B). The extent and impact of between-study heterogeneity was assessed by inspecting the forest plots and by calculating the tau^2^ (absolute heterogeneity) or the I^2^ statistics (relative heterogeneity). I^2^ defines the proportion of total variability in the result explained by heterogeneity, and not chance, while also considering the heterogeneity’s direction (localization on the forest plot) and uncertainty around heterogeneity estimates [23]. The 95% random-effects predictive intervals were calculated to incorporate observed heterogeneity and predict expected results in a future treatment [24].

### 2.8. Additional Analyses and Risk of Bias across Studies

Possible sources of heterogeneity were a priori planned to be sought through random-effects subgroup analyses and meta-regressions in meta-analyses of at least five trials, according to the following factors: appliance type (removable or fixed), patient age, patient sex, and treatment duration. Reporting biases were assessed with contour-enhanced funnel plots and Egger’s test [25] for meta-analyses with ≥10 studies.

The overall quality of meta-evidence (i.e., the strength of clinical recommendations) was rated using the Grades of Recommendations, Assessment, Development and Evaluation (GRADE) approach [18] following recent guidance for nonrandomized studies [26]. The produced forest plots were augmented with contours denoting the magnitude of the observed effects (Appendix B) to assess heterogeneity, clinical relevance, and imprecision [17].

Robustness of the results was checked for meta-analyses ≥ 5 studies with sensitivity analyses based on (i) the inclusion of prospective versus retrospective studies, (ii) unequal duration of treatment/observation between treated/control groups, (iii) inadequate matching (assessed with Cohen’s d for baseline measurements of each outcome), and (iv) studies with inadequate versus inadequate samples, with the cut-off set at 25 patients/group. All analyses were run in Stata version 14.0 (StataCorp LP, College Station, TX, USA) by one author (S.N.P.) and the dataset was openly provided [27]. All *p* values were two sided with α = 5%, except for the test of between-studies or between-subgroups heterogeneity where α-value was set as 10% [28].

## 3. Results

### 3.1. Study Selection

A total of 2095 hits were retrieved by the literature database search and another 6 records were identified manually (Figure 1). After removing duplicates and eliminating nonrelevant reports by title/abstracts, 185 full-text papers were checked against the eligibility criteria (Appendix A). In the end, 20 publications pertaining to 20 unique studies were included in this review.

### 3.2. Study Characteristics

All 20 included studies [29,30,31,32,33,34,35,36,37,38,39,40,41,42,43,44,45,46,47,48] were nonrandomized (Table 1), with only 4 studies (20%) being prospective. All studies were conducted within a university setting (one jointly with a hospital) in 9 different countries (Brazil, Egypt, India, Italy, Pakistan, Spain, Sweden, Turkey, and the United States of America). The included studies were all published as journal papers and were in English, except from one study that was in Turkish.

The eligible studies included a total of 969 patients (536 treated/433 untreated), to a median sample size of 40.5 patients/study (range: 20–93 patients/study). Among the 20 studies reporting the patients’ gender, 47.9% of the patients were male (424 of the total 886), while from the 16 studies reporting mean age, the average across studies was 10.9 years (range of average age/study 8.4–14.5 years). The identified studies used dental Class II molar relationship, cephalometric skeletal anteroposterior jaw relationship, overjet, or vertical jaw configuration as eligibility criteria to include patients, while 5 studies (25%) also included explicit reporting of no respiratory problems (Appendix A). Removable functional appliances (Activator, Fränkel-2, Twin Block, or Sander appliance) were used in 16 studies, while fixed functional appliances (Herbst, Forsus Fatigue Resistant Device, Mandibular Protraction Appliance-IV, Mandibular Anterior Repositioning Appliance, or X-Bow appliance) were used in 8 studies (with 4 studies using both removable and fixed appliances). One study [32] also included a prefabricated myofunctional appliance (Trainer 4 Kids), but this was omitted from the review, due to the different modus operandi [49]. One study [44] incorporated headgear to the Activator for anchorage reinforcement, while another study [31] also included a second phase treatment with braces after a first phase with Twin Block. Airway dimensions were assessed by lateral cephalograms in 16 (80%) of the studies and by CBCT in the remaining 4 (20%)—all of them made in an upright position. All studies reported outcome results before and after treatment with functional appliances, while only one study [35] reported long-term follow-up after treatment (6 years).

### 3.3. Risk of Bias within Studies

The risk of bias of included nonrandomized studies is summarized in Table 2 and given in detail in Appendix A. For most studies, inclusion of patients in the study was not based on any factor that could influence treatment outcome (85%), and the treatment/control groups were clearly defined (95%). Treated/untreated patients were explicitly reported to be selected from the same source and time in only half (50%) of the studies, while the rate of adequate matching at baseline for potential confounders (age, sex, malocclusion, airway dimensions, and treatment/observation duration) between treated/control patients ranged from 35% to 65%. Finally, no study blinded the person measuring the cephalometric/CBCT variables, while the sample size was deemed to be adequate (≥50 patients) in 4 (20%) studies. All included studies were judged to be in critical risk of bias, as issues existed for at least three domains per study.

### 3.4. Results of Individual Studies and Data Synthesis

The results of studies not included in any meta-analyses are given in Appendix A. Functional appliances were associated with a statistically significant but clinically irrelevant reduction in hypopharynx dimensions compared to untreated controls. Additionally, functional appliances were associated with statistically significant increases in nasopharynx dimensions, oropharynx cross-section, and pharynx height—with the increase in nasopharynx being also clinically relevant.

Meta-analyses of the effects of functional appliances on upper airway dimensions are given in Table 3. Orthopedic therapy with functional appliances was associated with statistically significant increases in the volume of both the nasopharynx (3 studies; SMD = 0.95; 95% CI = 0.36 to 1.54; *p* = 0.002) and the oropharynx (4 studies; MD = 2356.14 mm^3^; 95% CI = 1276.36 to 3435.92 mm^3^; *p* < 0.001; Figure 2) compared to natural growth.

Moderate heterogeneity existed among studies (I^2^ 60% and 69%, respectively), which, however, influenced only the precise quantification of the improvement seen through treatment (as all included studies were on the same side of the forest plot).

Furthermore, functional appliance therapy was associated with statistically significant increases in (i) the minimal constricted axial area of pharyngeal airway (2 studies; MD = 59.91 mm^2^); (ii) superoposterior airway space (8 studies; MD = 1.63 mm); (iii) middle airway space (11 studies; MD = 1.25 mm); (iv) inferior airway space (10 studies; MD = 1.32 mm); (v) McNamara’s lower pharynx dimension (3 studies; MD = 2.31 mm); (vi) lower adenoid thickness (2 studies; MD = 1.16 mm); and (vii) pharyngeal dimension at the epiglottal base (4 studies; MD = 0.70 mm). Heterogeneity was relatively moderate, except from the meta-analyses of middle and inferior airway space, where considerable heterogeneity was seen (I^2^ > 75%).

### 3.5. Subgroup and Meta-Regression Analyses

Differences in the effects of removable and fixed functional appliances were assessed in Table 4 and tested formally with subgroup interaction for meta-analyses of at least 5 studies. For most outcomes, removable functional appliances showed considerable greater benefits in terms of airway dimensions than fixed appliances, like nasopharynx volume (SMDs of 1.64 and 0.73, respectively), superoposterior airway space (MDs of 1.41 and 1.08 mm, respectively), middle airway space (MDs of 1.37 and 1.02 mm, respectively), and inferior airway space (MDs of 1.52 and 0.79 mm, respectively). Additionally, increases in McNamara’s lower pharynx and sagittal depth of the nasopharynx were seen only with removable functional appliances and had no significant effect with fixed functional appliances. Furthermore, an effect reversal was seen for the minimal constricted axial area of pharyngeal airway, where an increase was seen with removable appliances and a reduction was seen with fixed appliances. However, all these differences were not confirmed by formal subgroup interaction—possibly due to low statistical power. The only exception was for the primary outcome of oropharynx volume, where removable appliances induced a statistically significantly greater increase than fixed appliances (MDs of 2595.56 and 2303.57 mm^3^; *p* = 0.04).

Meta-regression analyses indicated that patient age had a significant influence on the effects of functional appliances (Table 5), as treatment-induced increase in posterior airway space was reduced on average by −0.36 mm (95% CI = −0.75 to 0.03 mm) for each additional year of age. Additionally, a dose-response relationship was seen between increases in airway and treatment duration, as for each additional treatment month, additional increases in superoposterior airway space (coefficient = 0.12 mm; 95% CI = −0.02 to 0.26 mm) and inferior airway space (coefficient = 0.29 mm; 95% CI = 0.12 to 0.45 mm) were seen. Limiting the meta-regressions to only removable functional appliances revealed a greater influence of patient’s age on increases in posterior airway space (coefficient = −0.99 mm; 95% CI = −1.91 to −0.08 mm) and of treatment duration on inferior airway space (coefficient = 0.41 mm; 95% CI = 0.26 to 0.56 mm), which might be anticipated, due to the greater treatment effects of removable appliances on the airways.

### 3.6. Reporting Biases and Sensitivity Analyses

Reporting biases (including the possibility for publication bias) could be assessed only for the meta-analyses of middle and inferior airway space that included at least 10 studies. The funnel plots (Appendix A) indicated asymmetry, which was confirmed by Egger’s test in both instances (*p* = 0.07 and *p* = 0.006, respectively). However, this was interpreted as small-study effects, with smaller/more imprecise studies reporting greater treatment effects than larger studies.

Sensitivity analyses according to methodological issues of existing studies are seen in Table 6. No significant differences in the meta-analyses were seen between prospective versus retrospective studies nor according to the difference between treatment and observation durations. However, adequate baseline matching had a significant effect on the reported treatment effects of functional appliances. Studies with greater baseline differences between treated/untreated patients (i.e., without adequate matching) reported significantly greater increases in middle (coefficient = 0.93 mm) and posterior airway space (coefficient = 1.86 mm). Furthermore, studies with adequate sample size (≥50 patients) reported significantly higher increases of inferior airway space (coefficient = 2.05 mm) compared to smaller studies. Therefore, future clinical recommendations should be based on studies with adequate baseline matching (preferably through randomization) and with adequate sample size.

### 3.7. Quality of Evidence

The quality of evidence according to GRADE was very low in all instances (downgraded by two points), due to the lack of randomization and the many methodological issues from the identified retrospective studies that might introduce bias. Therefore, our confidence in current estimates is very low and future studies might change current recommendations. 

## 4. Discussion

### 4.1. Evidence in Context

The present systematic review summarizes clinical evidence from existing studies assessing the effects of Class II orthopedic treatment with functional appliances on airway dimensions to untreated Class II controls. A total of 20 studies including 536 treated and 433 untreated Class II patients were finally included in the meta-analyses.

Mandibular advancement with removable or fixed functional appliances was associated with statistically significant increases in airway dimensions directly after treatment compared to what could be expected by Class II growth alone. Specifically, benefits were seen for volume of the naso- and oropharynx, the minimal constricted axial area of pharyngeal airway, and many sagittal measurements of the oropharynx (Table 3). However, many of these changes, especially at the upper pharynx, were small to moderate in magnitude, which means that they might have little clinical relevance. On the contrary, greater effects were seen at the lower part of the pharynx and this indicates that any clinically relevant benefits in airway dimensions or breathing might be attributed in this compartment. There is some evidence indicating that normal patients and patients with sleep-disordered breathing have significant differences in the dimensions of the pharyngeal airway or the thickness of the pharyngeal wall [50], and the lower retropalatal/retroglossal areas are mostly affected [51]. This area has also emerged as a sensitive parameter enabling to consistently assess the patient’s respiratory conditions [52]. However, even though increases in airway volume or cross-section might be indicative of improved breathing, functional confirmation through improved nasal airflow resistance, nasal pressure, and patient-relevant outcomes is needed. Some data indicate that treatment with Herbst appliance improved nocturnal breathing in adolescents [53], but the evidence is weak due to the lack of a control group and further studies are needed.

The exact mechanism with which these changes on the upper airway occur is currently unknown, but it might be that the mesial displacement of the lower dentition and the labial flaring of the lower incisors, might cause anterior traction on the tongue and hyoid bone [48], thereby causing adaptive changes of the soft palate and leading to an increase in pharyngeal airway dimensions [33]. This is also compatible with the observation that the soft palate is anteriorly repositioned after functional appliance treatment of Class II [39,41] as the tongue moves away from the palate. However, confirmatory studies are needed.

The effects of orthopedic mandibular advancement on the airways were highly variable among the included studies, which was reflected in between-study heterogeneity. Part of this heterogeneity was explained by several patient- or treatment-related characteristics including patient age, appliance design, and treatment duration (Table 4 and Table 5). Removable functional appliances were shown to exert greater changes in the upper airway dimensions than fixed appliances for most of the analyzed variables. This might be due to different skeletal/dentoalveolar effects of removable/fixed appliances that have been previously reported [4]. On the other hand, this might be due to the fact that fixed functional appliances are usually placed on older patients after most deciduous teeth have been shed, whereas removable functional appliances are often placed in the mixed dentition (an age difference also seen among the included studies). This might act as a confounding factor at least to some part, since patient age was consistently associated with the observed airway benefits, both for the whole set of included studies and, specifically, for the subset of removable appliances (Table 5). It is generally believed that, moreover, skeletal effects of functional appliances are more pronounced in patients treated before or during the growth peak [54].

Existing clinical studies only demonstrated the short-term beneficial effect of functional appliances on the upper airways. However, it remains to be seen whether such benefits remain stable in the long term. The sole included study assessing long-term status of treated Class II patients [35] indicated that not only were the benefits of functional appliance treatment retained 6 years afterwards, but a significantly greater post-treatment increase was seen. This is also consistent with previous evidence on the long-term stability of increased airway dimensions among patients with skeletal Class III treated orthopedically with maxillary protraction [55].

### 4.2. Strengths and Limitations

Among the strengths of the current review can be counted it is a priori registration [15], the extensive searching of the literature, the inclusion of untreated Class II controls, the use of contemporary statistical methods [22], the gauging of the quality of meta-evidence according to GRADE [18], and the transparent open provision of the dataset [56].

On the other hand, some limitations also exist, like the inclusion of weak study designs like retrospective nonrandomized studies [57] with historical controls [58], which might introduce bias. Additionally, most studies had small sample sizes and this can affect the precision of the estimated effects [59]. Moreover, many included studies were inadequately matched in terms of similar baseline airway dimensions, and baseline dissimilarities were associated with inflated treatment effects (Table 6), which is in agreement with previous meta-epidemiological evidence [60,61]. Furthermore, airways before and after treatment were assessed with radiographs done in the upright position and not in a supine position, since most studies were retrospective with nonapneic patients that received functional appliance treatment for their underlying malocclusion and airways were only secondarily assessed. However, changes between supine or upright posture can influence airways measurements [62,63,64], even though oropharyngeal area measurement from lateral cephalograms can be used as an initial screening measurement to predict the upright upper airway 3D volume [64]. Finally, the small number and the limited reporting of existing studies did not enable extensive subgroup and meta-regression analyses to identify and account for sources of confounding, like patient age, sex, growth pattern, and presence/size of tonsils or adenoid, which might influence the observed results. Further prospective, ideally randomized, studies with open provision of their full dataset [56] will help in shedding on the pure airway effects of orthopedic treatment for Class II malocclusion.

## 5. Conclusions

Current evidence indicates that orthopedic treatment with functional appliances for Class II malocclusion might be associated with increased volume and dimensions of the upper airways, which are dependent on patient- and treatment-related factors. However, our confidence in these data is very low due to the poor quality of existing studies and their small number. It is crucial that the clinical relevance of such anatomical changes is confirmed by functional analyses of breathing ability before concrete recommendations can be formulated.

## Figures and Tables

**Figure 1 jcm-09-03806-f001:**
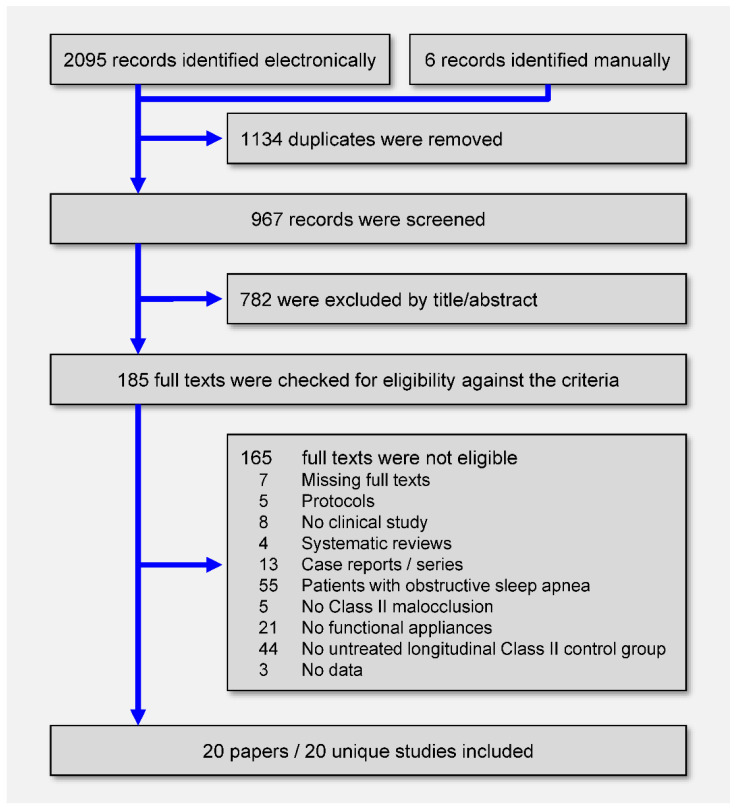
Preferred Reporting Items for Systematic Reviews and Meta-Analyses (PRISMA) flow diagram for the identification and selection of studies.

**Figure 2 jcm-09-03806-f002:**
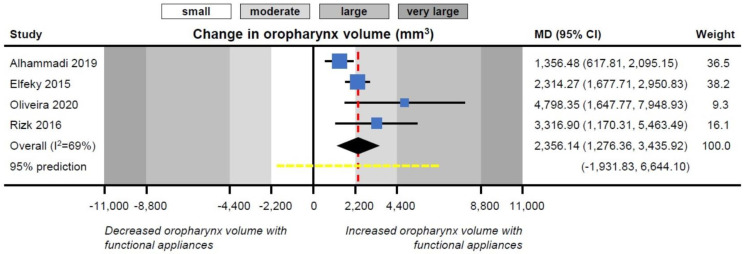
Contour-enhanced forest plot for the effect of functional appliances on oropharynx volume.

**Table 1 jcm-09-03806-t001:** Characteristics of included studies.

Study	Design; Setting; Country *	Patients (M/F); Age ^†^	Appliance (Active Duration)	Radio-Graph
Aksu 2017 [29]	rNRS; Uni; TR	Exp: 16 (4/12); 10.3 Control: 19 (9/10); 10.2	Exp: Activator (15.6) Control—(12.0)	Lateral ceph
Alhammadi 2019 [30]	pNRS; Uni; EG	Exp1: 23 (0/23); 11.9 Exp2: 21 (0/21); 13.5 Control: 18 (0/18); 11.3	Exp1: Twin Block (Tx end) Exp2: Forsus FRD (Tx end) Control—(as Exp1–2)	CBCT
Ali 2015 [31]	rNRS; Uni; PK	Exp: 42 (21/21); 10.4 Control ^‡^: 32 (16/16); 10.1	Exp: Twin Block +FA (36.4) Control—(36.0)	Lateral ceph
Atik 2017 [32]	rNRS; Uni; TR	Exp1: 15 (4/11); 8.9 Exp2: 15 (6/9); 10.6 Control: 10 (6/4); 9.3	Exp1: Fränkel-2 (14.3) Exp2: X-Bow (8.6) Control—(14.8)	Lateral ceph
Bavbek 2016 [33]	rNRS; Uni; TR	Exp: 18 (10/8); 13.6 Control: 19 (8/11); 12.7	Exp: Forsus FRD (8.7) Control—(11.9)	Lateral ceph
Cortese 2020 [34]	rNRS; Uni; IT	Exp: 10 (7/3); 10.9 Control: 10 (5/5); 10.1	Exp: Activator/Twin Block (21.6) Control—(40.8)	Lateral ceph
Drosen 2018 [35]	rNRS; Uni; SE	Exp: 13 (13/0); 12.4 Control ^‡^: 13 (13/0); 12.1	Exp: Herbst (21.6) Control—(25.2)	Lateral ceph
Elfeky 2015 [36]	pNRS; Uni; EG	Exp: 18 (0/18); 10.0–12.0 Control: 18 (0/18); 10.0–12.0	Exp: Twin Block (8.0) Control—(8.0)	CBCT
Entrenas 2019 [37]	pNRS; Uni; ES	Exp: 40 (20/20); 9.8 Control: 20 (10/10); 9.1	Exp: Twin Block (Tx end) Control—(12.0–24.0)	Lateral ceph
Fabiani 2017 [38]	rNRS; Uni; IT	Exp: 28 (13/15); 8.4 Control ^‡^: 21 (14/7); 8.5	Exp: Fränkel-2 (14.6) Control—(16.0)	Lateral ceph
Ghodke 2014 [39]	pNRS; Uni; IN	Exp: 20 (11/9); 8.0–13.0 Control: 18 (9/9); 8.0–14.0	Exp: Twin Block (6.0) Control: ± sectionals (6.0)	Lateral ceph
Goymen 2019 [40]	rNRS; Uni, TR	Exp1: 15 (7/8); 12.1 Exp2: 15 (7/8); 14.5 Control ^‡^: 10 (NR); 13.0	Exp1: Twin Block (Tx end) Exp2: Forsus FRD (Tx end) Control—(6.0)	Lateral ceph
Jena 2013 [41]	rNRS; Uni; IN	Exp1: 16 (9/7); 12.8 Exp2: 21 (11/10); 11.4 Control 16 (9/7); 10.6	Exp1: MAPA4 (6.2) Exp2: Twin Block (9.4) Control: ± sectionals (9.9)	Lateral ceph
Kilinc 2018 [42]	uNRS; Uni; TR	Exp: 19 (11/8); NR Control: 19 (7/12); NR	Exp: Activator (11.5) Control—(11.3)	Lateral ceph
Oliveira 2020 [43]	rNRS; Uni; BR	Exp: 24 (15/9); NR Control: 18 (10/8); NR	Exp: Herbst (8.0) Control: Pre-Tx (10.4)	CBCT
Ozbek 1998 [44]	rNRS; Uni; TR	Exp: 26 (11/15) 11.5 Control: 15 (7/8) 11.3	Exp: Activator±headgear (17.4) Control—(23.0)	Lateral ceph
Pavoni 2017 [45]	uNRS; Uni; IT	Exp: 51 (27/24); 9.9 Control: 31 (15/16); 10.1	Exp: Activator (21.6) Control—(22.8)	Lateral ceph
Rizk 2016 [46]	rNRS; Uni; US	Exp: 20 (7/13); 11.7 Control: 73 (NR); NR	Exp: MARA+FA (27.4) Control—(NR)	CBCT
Rongo 2020 [47]	rNRS; Hosp/Uni; IT	Exp: 34 (21/13); 11.1 Control: 34 (25/9); 10.4	Exp: Sander (14.8) Control—(13.9)	Lateral ceph
Ulusoy 2014 [48]	rNRS; Uni; TR	Exp: 16 (8/8); 11.4 Control: 19 (8/11); 12.1	Exp: Activator (11.0) Control—(11.4)	Lateral ceph

* given with the country’s ISO 3166 alpha-2 code, ^†^ given as mean (one value) or if mean not reported, given as range (two values), ^‡^ historical control from growth study or archive. CBCT, cone beam computerized tomography; ceph, cephalogram; Exp, experimental group; FA, fixed appliance (braces); FRD, fatigue-resistant device; MAPA4, Mandibular Protraction Appliance-IV; MARA, Mandibular Anterior Repositioning Appliance; pNRS, prospective nonrandomized study; Pract, private practice; rNRS, retrospective nonrandomized study; Tx, treatment; Uni, university clinic; uNRS; nonrandomized study with unclear design.

**Table 2 jcm-09-03806-t002:** Risk of bias summary of included nonrandomized studies.

Question	Yes/Probably Yes	No/Probably No	No Information
Was the study prospective?	5 (25%)	15 (75%)	–
Was selection of patients based on any factor that could influence the outcome (malocclusion, airways, compliance, missed appointments, breakages)?	3 (15%)	17 (85%)	–
Were FA/CTR groups clearly defined?	19 (95%)	1 (5%)	–
Were FA/CTR patients treated/observed at the same place/time?	10 (50%)	4 (20%)	6 (30%)
Were FA/CTR patients matched for baseline age?	11 (55%)	5 (25%)	4 (20%)
Were FA/CTR patients matched for baseline sex?	13 (65%)	5 (25%)	2 (10%)
Were FA/CTR patients matched for baseline malocclusion?	12 (60%)	5 (25%)	3 (15%)
Were FA/CTR patients matched for baseline airway measurements?	7 (35%)	13 (65%)	–
Was the use of other appliances the same among FA/CTR patients?	14 (70%)	6 (30%)	–
Was the observation period similar for FA/CTR patients?	9 (45%)	7 (35%)	4 (20%)
Were FA/CTR patients measured exactly the same way?	20 (100%)	–	–
Were FA/CTR patients measured blindly?	–	20 (100%)	–
Was the adequate sample? (25 patients per group)	4 (20%)	16 (80%)	–

CTR, untreated control group; FA, functional appliance group.

**Table 3 jcm-09-03806-t003:** Random-effects meta-analyses for the effect of any functional appliance versus untreated controls on airways.

Outcome	Studies	MD (95% CI)	*p*	I^2^ (95% CI)	tau^2^ (95% CI)	95% Prediction
Superoposterior airway space (mm)	8	1.63 (1.03, 2.23)	<0.001	68% (28%, 92%)	0.42 (0.08, 2.15)	−0.13, 3.39
Posterior airway space (mm)	8	0.52 (−0.20, 1.24)	0.15	47% (0%, 87%)	0.44 (0, 3.44)	−1.34, 2.38
Middle airway space (mm)	11	1.25 (0.53, 1.98)	0.001	81% (58%, 93%)	1.09 (0.36, 3.69)	−1.25, 3.76
Inferior airway space (mm)	10	1.32 (0.34, 2.31)	0.009	90% (76%, 97%)	1.97 (0.75, 6.47)	−2.12, 4.76
McNamara’s upper pharynx (mm)	3	1.35 (−0.57, 3.27)	0.17	87% (45%, 99%)	2.45 (0.31, 48.50)	−22.12, 24.82
McNamara’s lower pharynx (mm)	3	2.31 (0.79, 3.82)	0.003	70% (0%, 99%)	1.18 (0, 41.05)	−14.64, 19.25
Upper adenoid thickness (AD2-H; mm)	2	0.24 (−2.10, 2.58)	0.84	93% (NE)	2.65 (NE)	NE
Lower adenoid thickness (AD1-Ba; mm)	2	1.16 (0.46, 1.86)	0.001	0% (NE)	0 (NE)	NE
Upper airway thickness (PNS-AD2; mm)	5	0.38 (−0.18, 0.94)	0.19	13% (0%, 89%)	0.06 (0, 3.00)	−0.81, 1.57
Nasopharynx height (PNS-BaN; mm)	2	0.13 (−0.77, 1.02)	0.78	51% (NE)	0.21 (NE))	NE
Upper pharyngeal airway passage (Ptm-UPW; mm)	2	−0.37 (−1.73, 0.99)	0.60	0% (NE)	0 (NE)	NE
Base of epiglottis-posterior pharyngeal wall (V-LPW; mm)	4	0.70 (0.11, 1.29)	0.02	14% (0%, 93%)	0.05 (0, 4.46)	−0.93, 2.33
Sagittal depth of bony nasopharynx (Ba-PNS; mm)	2	1.25 (0.06, 2.43)	0.04	21% (NE)	0.18 (NE)	NE
Minimum axial area (mm^2^)	2	59.91 (41.46, 78.35)	<0.001	0% (NE)	0 (NE)	NE
Oropharynx sagittal dimension (mm)	2	1.20 (−2.12, 4.52)	0.48	97% (80%, 100%)	5.58 (0.68, 721.82)	NE
Oropharynx area (units)	2 *	556.10 (−279.88, 1392.08)	0.19	0% (NE)	0 (NE)	NE
Nasopharynx volume (mm^3^)	3	0.95 ^†^ (0.36, 1.54)	0.002	60% (0%, 98%)	0.16 (0, 5.02)	−5.44, 7.34
Oropharynx volume (mm^3^)	4	2356.14 (1276.36, 3435.92)	<0.001	69% (0%, 98%)	>100 (0, >100)	−1931.83, 6644.10

CI, confidence interval; MD, mean difference; NE, not estimable, * Study of Ozbek 1998 omitted due to different measurement method, ^†^ SMD used instead of MD due to big differences in the control group baseline measurements.

**Table 4 jcm-09-03806-t004:** Subgroup analyses for the effect of removable or fixed functional appliances analyses versus untreated controls on airways.

	All Appliances		Removable Appliances		Fixed Appliances		
Outcome	MD (95% CI)	*p*	MD (95% CI)	*p*	MD (95% CI)	*p*	P_SG_
Superoposterior airway space (mm)	*n* = 8 1.63 (1.03, 2.23)	<0.001	*n* = 7 1.41 (0.65, 2.17)	<0.001	*n* = 2 1.08 (0.35, 1.82)	0.004	0.20
Posterior airway space (mm)	*n* = 8 0.52 (−0.20, 1.24)	0.15	*n* = 6 0.83 (−0.18, 1.84)	0.11	*n* = 2 0.14 (−0.77, 1.06)	0.76	0.35
Middle airway space (mm)	*n* = 11 1.25 0.53, 1.98)	0.001	*n* = 9 1.37 (0.47, 2.26)	0.003	*n* = 3 1.02 (0.29, 1.75)	0.006	0.26
Inferior airway space (mm)	*n* = 10 1.32 (0.34, 2.31)	0.009	*n* = 8 1.52 (0.32, 2.72)	0.01	*n* = 2 0.79 (0.03, 1.54)	0.04	0.15
McNamara’s upper pharynx (mm)	*n* = 3 1.35 (−0.57, 3.27)	0.17	*n* = 2 2.05 (−0.04, 4.14)	0.06	*n* = 1 −0.20 (−1.81, 1.41)	0.81	NT
McNamara’s lower pharynx (mm)	*n* = 3 2.31 (0.79, 3.82)	0.003	*n* = 2 2.95 (2.13, 3.78)	<0.001	*n* = 1 0 (−2.67, 2.67)	1.00	NT
Upper adenoid thickness (AD2-H; mm)	*n* = 2 0.24 (−2.10, 2.58)	0.84	*n* = 2 0.24 (−2.10, 2.58)	0.84	−		NT
Lower adenoid thickness (AD1-Ba; mm)	*n* = 2 1.16 (0.46, 1.86)	0.001	*n* = 2 1.16 (0.46, 1.86)	0.001	−		NT
Upper airway thickness (PNS-AD2; mm)	*n* = 5 0.38 (−0.18, 0.94)	0.19	*n* = 4 0.13 (−0.51, 0.78)	0.69	*n* = 1 0.61 (−1.90, 3.12)	0.63	0.73
Nasopharynx height (PNS-BaN; mm)	*n* = 2 0.13 (−0.77, 1.02)	0.78	*n* = 2 0.27 (−1.01, 1.56)	0.68	*n* = 1 0.02 (−0.88, 0.92)	0.97	NT
Upper pharyngeal airway passage (Ptm-UPW; mm)	*n* = 2 −0.37 (−1.73, 0.99)	0.60	n = 2 −0.04 (−1.52, 1.44)	0.96	*n* = 1 −1.12 (−3.06, 0.82)	0.26	NT
Base of epiglottis-posterior pharyngeal wall (V-LPW; mm)	*n* = 4 0.70 (0.11, 1.29)	0.02	*n* = 3 0.65 (−0.33, 1.62)	0.19	*n* = 2 0.51 (−0.46, 1.48)	0.30	NT
Sagittal depth of bony nasopharynx (Ba-PNS; mm)	*n* = 2 1.25 (0.06, 2.43)	0.04	*n* = 2 1.62 (0.57, 2.68)	0.003	*n* = 1 −0.71 (−2.91, 1.49)	0.53	NT
Minimum axial area (mm^2^)	*n* = 2 59.91(41.46, 78.35)	<0.001	*n* = 2 91.60 (19.14, 197.56)	0.01	*n* = 1 −26.97 (−44.18, −9.76)	0.002	NT
Oropharynx sagittal dimension (mm)	*n* = 2 1.20 (−2.12, 4.52)	0.48	*n* = 1 −0.65 (−0.89, −0.42)	<0.001	*n* = 2 1.30 (−1.83, 4.42)	0.42	NT
Oropharynx area (units)	*n* = 2 * 556.10 (−279.88, 1392.08)	0.19	*n* = 2 114.35 (98.61, 130.09)	<0.001	*n* = 1 607.00 (−452.17, 1666.17)	0.26	NT
Nasopharynx volume (mm^3^)	*n* = 3 0.95 ^†^ (0.36, 1.54)	0.002	*n* = 1 1.64 ^†^ (0.88, 2.40)	<0.001	*n* = 1 0.73 ^†^ (0.10, 1.36)	0.02	NT
Oropharynx volume (mm^3^)	*n* = 4 2356.14 (1276.36, 3435.92)	<0.001	*n* = 2 2595.56 (2013.07, 3178.05)	<0.001	*n* = 3 2303.57 (−808.11, 5415.25)	0.15	0.04

CI, confidence interval; MD, mean difference; NT, not tested; P_SG_, *p* value for subgroup differences. * Study of Ozbek 1998 omitted due to different measurement method, ^†^ SMD used instead of MD due to big differences in the control group baseline measurements.

**Table 5 jcm-09-03806-t005:** Meta-regression analysis on the effect of functional appliances on airways.

	Any Functional Appliance (Removable/Fixed)			Only Removable Appliances		
Outcome	Patient Age (Per Year)	Male % in Sample (Per %)	Treatment Duration (Per Month)	Patient Age (Per Year)	Male % in Sample (Per %)	Treatment Duration (Per Month)
Upper airway thickness (PNS-AD2; mm)	b = −0.55 *p* = 0.30	b = −4.03 *p* = 0.43	b = 0.03 *p* = 0.26	NT	b = −4.71 *p* = 0.46	NT
Superoposterior airway space (mm)	b = 0.11 *p* = 0.69	b = 1.99 *p* = 0.47	b = 0.12 *p* = 0.09	b = 0.41 *p* = 0.41	b = 2.13 *p* = 0.49	b = 0.17 *p* = 0.12
Posterior airway space (mm)	b = −0.36 *p* = 0.06	b = −6.00 *p* = 0.15	b = 0.07 *p* = 0.45	b = −0.99 *p* = 0.04	b = −6.15 *p* = 0.24	b = 0.07 *p* = 0.70
Middle airway space (mm)	b = 0.03 *p* = 0.91	b = −2.59 *p* = 0.41	b = 0.09 *p* = 0.28	b = 0.10 *p* = 0.84	b = −3.40 *p* = 0.37	b = 0.09 *p* = 0.48
Inferior airway space (mm)	b = −0.13 *p* = 0.78	b = 1.56 *p* = 0.71	b = 0.29 *p* = 0.003	b = 0.02 *p* = 0.98	b = 1.61 *p* = 0.73	b = 0.41 *p* < 0.001
Base of epiglottis-posterior pharyngeal wall (V-LPW; mm)	b = 0.29 *p* = 0.46	b = −12.09 *p* = 0.41	b = −0.13 *p* = 0.38	NT	b = −13.62 *p* = 0.55	NT
Oropharynx volume (mm^3^)	b = −1256.28 *p* = 0.17	b = 5027.43 *p* = 0.23	b = 13.37 *p* = 0.92	NT	NT	NT

b, meta-regression coefficient; NT, not tested (as less than 5 studies contributed to the analysis).

**Table 6 jcm-09-03806-t006:** Sensitivity analyses on the effect of methodological characteristics on the effect of functional appliances on airways.

Outcome	Prospective Vs Retrospective (Ref)	Tx-Ctr Difference in Duration (Per Month)	Tx-Ctr Difference in Baseline Outcome (In Absolute Cohen’s d)	Adequate Sample (≥50) vs. Inadequate (Ref)
Upper airway thickness (PNS-AD2; mm)	NE	b = 0.05 *p* = 0.82	b = 0.38 *p* = 0.26	b = 0.39 *p* = 0.59
Superoposterior airway space (mm)	NE	b = −0.07 *p* = 0.32	b = 0.82 *p* = 0.15	b = 0.91 *p* = 0.36
Posterior airway space (mm)	NE	b = 0.04 *p* = 0.83	b = 1.86 *p* = 0.08	b = −0.46 *p* = 0.65
Middle airway space (mm)	b = −0.65 *p* = 0.62	b = 0.01 *p* = 0.91	b = 0.93 *p* = 0.04	b = −1.56 *p* = 0.27
Inferior airway space (mm)	NE	b = −0.03 *p* = 0.79	b = 0.51 *p* = 0.64	b = 2.05 *p* = 0.09
Base of epiglottis-posterior pharyngeal wall (V-LPW; mm)	b = 0.95 *p* = 0.26	b = 0.03 *p* = 0.90	b = −0.48 *p* = 0.60	b = −1.45 *p* = 0.29
Oropharynx volume (mm^3^)	b = −2334.16 *p* = 0.27	NE	b = −3586.53 *p* = 0.36	b = −1406.38 *p* = 0.51

b, meta-regression coefficient; Ctr, control; NE, not estimable; Ref, reference; Tx, treatment.

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
