# Peer review of "Orthopedic Treatment for Class II Malocclusion with Functional Appliances and Its Effect on Upper Airways: A Systematic Review with Meta-Analysis"

_jcm, 2020, doi:10.3390/jcm9123806_

Round 1

Reviewer 1 Report

A very well-done conducted study about a hot topic in modern orthodontics. 

The selection of papers is well descripted and results obtained are well exposed Conclusions are coherent with results. 

Table 1 is a bit confusing: please modify it in order to make it more readble.

Author Response

REVIEWER #1

Comment #1

“A very well-done conducted study about a hot topic in modern orthodontics”.

Response #1

We thank the reviewer for this supportive comment.

Changes #1

None.

Comment #2

“The selection of papers is well descripted and results obtained are well exposed Conclusions are coherent with results”.

Response #2

Thank you.

Changes #2

None.

Comment #3

Table 1 is a bit confusing: please modify it in order to make it more readble.

Response #3

Thank you for this constructive comment. We have taken that into consideration and have split Table 1 in two parts: (i) one remaining as Table 1, but with much less abbreviations so that it is readable and (ii) and the remaining being inserted as a new Appendix S4, again with no abbreviations so that it’s easily readable.

Changes #3

See revised Table 1 and newly-added Appendix S4 (the following Appendices have been re-numbered).

Reviewer 2 Report

The article was written with a correct scientific methodology

The introduction section provide relevant and sufficient references. It explain clearly what is already known about the effect  produced by orthopedic treatment of class II patients on the airways.  I suggest to the authors to add the following references in the introduction section about the increase of the upper airways after orthopedic treatment https://doi.org/10.3390/ma13102239

The present systematic review is properly conducted with a priori registration on PROSPERO, and it is guided by the Cochrane Handbook and the PRISMA statement.

A very proper statistical analysis has been performed by the authors.

The data are presented in an appropriate way. Tables and figures are presented in the results section relevant and clearly Titles, columns, and rows are labelled correctly and clearly. The authors described precisely which variable was statistically significant

The discussione clearly explain the results.

The conclusion section answer the aim of the study and it is supported by the results of the study. Limitations of the study are opportunities for future authors to perform new reasearches about the effects of orthopedic class II treatment on the airways exceeding the limitations of the studies that have been included in this systematic review such as: small sample size, inadequate sample selection etc.

Author Response

REVIEWER #2

Comment #1

“The article was written with a correct scientific methodology”

Response #1

Thank you for this positive comment.

Changes #1

None.

Comment #2

“The introduction section provide relevant and sufficient references. It explain clearly what is already known about the effect produced by orthopedic treatment of class II patients on the airways. I suggest to the authors to add the following references in the introduction section about the increase of the upper airways after orthopedic treatment https://doi.org/10.3390/ma13102239”

Response #2

Thank you for the positive comment and the helpful suggestion, which we have adopted.

Changes #2

See revised Reference #12.

Comment #3

“The present systematic review is properly conducted with a priori registration on PROSPERO, and it is guided by the Cochrane Handbook and the PRISMA statement”.

Response #3

Thank you for this comment.

Changes #3

None.

Comment #4

“A very proper statistical analysis has been performed by the authors”

Response #4

Thank you.

 Changes #4

None.

Comment #5

“The data are presented in an appropriate way. Tables and figures are presented in the results section relevant and clearly Titles, columns, and rows are labelled correctly and clearly. The authors described precisely which variable was statistically significant”

Response #5

Thank you for these nice words.

Changes #5

None.

Comment #6

“The discussione clearly explain the results”.

Response #6

Thank you for this acknowledgement.

 Changes #6

None.

Comment #7

“The conclusion section answer the aim of the study and it is supported by the results of the study. Limitations of the study are opportunities for future authors to perform new reasearches about the effects of orthopedic class II treatment on the airways exceeding the limitations of the studies that have been included in this systematic review such as: small sample size, inadequate sample selection etc”

Response #7

Thank you for this positive comment.

Changes #7

None.

Reviewer 3 Report

The purpose of this systematic review was to compare the effects of Class II functional appliance on the change of the upper airway dimensions with natural occurring growth in untreated Class II patients.

Although the authors did a lot of efforts to perform this study, there are some major drawbacks.

  • If the authors want to report the real change in the upper airway, lateral cephalograms and CBCTs should be taken in the supine position. If not, what is the clinical meaning of the upper airway change? It should be clearly explained in the materials and method section and the discussion section.
  • The age range of patients is important for interpreting the results of upper airway change. If the age of patient is around 10-12 years, the upper airway can be changed by increase or decrease of the tonsil and adenoid, not by the pure effect of Class II functional appliance. Furthermore, how can the authors completely remove other confounding factors (for examples, difference in the mandibular growth between boy and girl, or between hyperdivergent and normo-or hypodivergent growth pattern) in untreated Class II patients. It should be clearly explained in the materials and method section and the discussion section.

Author Response

REVIEWER #3

Comment #1

“The purpose of this systematic review was to compare the effects of Class II functional appliance on the change of the upper airway dimensions with natural occurring growth in untreated Class II patients.

Although the authors did a lot of efforts to perform this study, there are some major drawbacks”

Response #1

We thank the reviewer for the time invested in this review. We have tried our best to take the provided comments into consideration and improve the quality of our manuscript.

Changes #1

See following comments.

Comment #2

“If the authors want to report the real change in the upper airway, lateral cephalograms and CBCTs should be taken in the supine position. If not, what is the clinical meaning of the upper airway change? It should be clearly explained in the materials and method section and the discussion section.”

Response #2

We thank the reviewer for pointing out this very valid point. As most of the studies were of retrospective nature and all of the studies included non-apneic patients where airway assessment was a secondarily outcome and not the reason for treatment, conventional upright radiographs were performed, since postural changes can have an effect on both airways and jaw relationship.

We have made this clear in the Materials & Methods section, in the Results section, and then discussed it further as a limitation in the Discussion section.

Changes #2

See revised text in the Methods section (page 2, line 81):

“Secondary outcomes included other measures of airway dimensions in terms of linear distances or areas, measured either on lateral cephalograms or CBCTs and in either upright or supine position.”

See revised text in the Results section (page 5, line 168):

“Airway dimensions were assessed by lateral cephalograms in 16 (80%) of the studies and by CBCT in the remaining 4 (20%) – all of them made in an upright position.”

See revised text in the Discussion section (page 8, line 317):

“Furthermore, airways before and after treatment were assessed with radiographs done in the upright position and not in a supine position, most studies were retrospective with non-apneic patients that received functional appliance treatment for their underlying malocclusion and airways were only secondarily assessed. However, changes between supine or upright posture can influence airways measurements [62-64], even though oropharyngeal area measurement from lateral cephalograms can be used as an initial screening measurement to predict the upright upper airway 3D volume [64].”

Comment #3

“The age range of patients is important for interpreting the results of upper airway change. If the age of patient is around 10-12 years, the upper airway can be changed by increase or decrease of the tonsil and adenoid, not by the pure effect of Class II functional appliance. Furthermore, how can the authors completely remove other confounding factors (for examples, difference in the mandibular growth between boy and girl, or between hyperdivergent and normo-or hypodivergent growth pattern) in untreated Class II patients. It should be clearly explained in the materials and method section and the discussion section.”

Response #3

We thank the reviewer for this considerate comment, which we have integrated in our revised manuscript.

Changes #3

See revised text in Discussion section (page 8, paragraph 323):

“Finally, the small number and the limited reporting of existing studies did not enable extensive subgroup and meta-regression analyses to identify and account for sources of confounding, like patient age, sex, growth pattern, and presence/size of tonsils or adenoid, which might influence the observed results. Further prospective, ideally randomized, studies with open provision of their full dataset [56] will help in shedding on the pure airway effects of orthopedic treatment for Class II malocclusion.”